# Circular RNAs Activity in the Leukemic Bone Marrow Microenvironment

**DOI:** 10.3390/ncrna8040050

**Published:** 2022-07-01

**Authors:** Francesca Liccardo, Alessia Iaiza, Martyna Śniegocka, Silvia Masciarelli, Francesco Fazi

**Affiliations:** Department of Anatomical, Histological, Forensic & Orthopedic Sciences, Section of Histology & Medical Embryology, Sapienza University of Rome, Via A. Scarpa, 14-16, 00161 Rome, Italy; francesca.liccardo@uniroma1.it (F.L.); alessia.iaiza@uniroma1.it (A.I.); martyna.sniegocka@uniroma1.it (M.Ś.)

**Keywords:** circRNA, acute myeloid leukemia, bone marrow niche, tumor microenvironment, non-coding RNA, hematopoietic stem cell, therapy resistance

## Abstract

Acute myeloid leukemia (AML) is a hematological malignancy originating from defective hematopoietic stem cells in the bone marrow. In spite of the recent approval of several molecular targeted therapies for AML treatment, disease recurrence remains an issue. Interestingly, increasing evidence has pointed out the relevance of bone marrow (BM) niche remodeling during leukemia onset and progression. Complex crosstalk between AML cells and microenvironment components shapes the leukemic BM niche, consequently affecting therapy responsiveness. Notably, circular RNAs are a new class of RNAs found to be relevant in AML progression and chemoresistance. In this review, we provided an overview of AML-driven niche remodeling. In particular, we analyzed the role of circRNAs and their possible contribution to cell–cell communication within the leukemic BM microenvironment. Understanding these mechanisms will help develop a more effective treatment for AML.

## 1. Introduction

Acute myeloid leukemia (AML) is the most common form of acute leukemia in adults. It is aggressive cancer characterized by clonal expansion and progressive accumulation of incompletely differentiated myeloid cells within the bone marrow. AML is a heterogeneous disease because of the plethora of mutations responsible for its development. Thus it is defined by diverse symptoms, prognosis and treatment responsiveness [1]. In most patients, etiology remains unknown, with the exception of therapy-related leukemias due to the previous exposure to chemotherapy and/or radiation administered for a primary condition. Incidence is higher in elderly people than in younger, and males are 1.2–1.6 times more likely to develop AML than females. Furthermore, conventional frontline treatments such as intensive chemotherapy and allogeneic stem cell transplantation are reliable and more effective in young and fit patients than in old/unfit ones [1,2,3]. Treatment is mostly based on conventional chemotherapy with a generally dismal outcome, especially in the elderly, with a 5-year survival of 30–35% in younger patients (age < 60 years) and 10–15% in older ones (age ≥ 60 years) [4]. This large gap in therapy efficacy is beginning to close thanks to the approval of nine new drugs for targeted therapy of AML within just three years (2017–2020). An enormous effort in better understanding of AML genomic and molecular landscape has made this possible [5]. Although these different compounds are promising targeted therapies for many distinct AML subtypes [6,7], still primary and secondary resistance remains an issue [8]. Therefore, there is an extreme need to explore the mechanisms of resistance to treatments. In the last few years, many new evidences stressed the role of the bone marrow niche in sustaining leukemic cells’ survival, progression and resistance to chemotherapy. The maintenance of this integrated system occurs through the exchange of factors and signals, generating intense cell–cell crosstalk. Numerous studies found several non-coding RNAs (ncRNAs), such as long non-coding RNAs, microRNAs and circular RNAs, playing a prominent role in both AML leukemogenesis and chemoresistance [9,10,11,12]. In this review, we aimed to report new insights into the interplay between AML cells and the microenvironment in order to provide a comprehensive view of the role of niche components during leukemic progression. Moreover, since the clinical significance of microRNA and long non-coding RNA has been investigated for several years, we highlighted the very last insights on the role of circular RNAs (circRNAs) in AML pathogenesis, with a particular focus on their function as potential mediators of crosstalk within the bone marrow niche.

## 2. Hematopoietic Stem Cells and Their Niche

Hematopoiesis is a process starting from a small group of Hematopoietic Stem Cells (HSCs) which are multipotent precursors endowed with self-renewal capacity and the ability to generate all types of mature blood cells through multi-lineage differentiation programs [13]. HSCs are identified on the basis of the expression of specific surface markers. Indeed, murine HSCs are identified by a marker combination Lin^neg/low^, Thy1.1^low^, c-Kit^high^, Sca-1^+^], and a similar combination is detected in human HSCs Lin^−^, Thy1^+^, CD34^+^, CD38^neg/low^] [14]. Moreover, HSCs can be divided on the basis of their differentiation potential. Long-term HSCs (LT-HSCs) have an indefinite self-renewal capacity, while their short-term derivative HSCs (ST-HSCs) maintain self-renewal property for eight weeks, then giving rise to multipotent progenitors (MMPs). HSCs are rare cells located in a complex and heterogeneous bone marrow microenvironment composed of both hematopoietic and non-hematopoietic cells surrounded by an extracellular matrix, in a ratio of 1:5000 for LT-HSC and 1:1000 for ST-HSC and multipotent progenitors in the murine bone marrow. Other components of the bone marrow are mesenchymal stem and progenitor cells, osteoprogenitor cells, perivascular stromal cells, endothelial cells, adipocytes, unmyelinated Schwann cells and cells of the immune system. The spatial organization and composition of these populations play a pivotal role in the regulation of HSCs’ maintenance and fate decisions. Although the specific location of HSCs still remains not fully clear, they preferentially localize in perisinusoidal areas [15,16]. Indeed, bone marrow is highly vascular, with arterioles close to the endosteum and sinusoids winding through a network of reticular stromal cells. Especially endothelial cells and perivascular stromal cells support HSCs maintenance and long-term repopulating activity by producing factors such as chemokine CXCL12, angiopoietin and stem cell factor (SCF) [17,18,19]. Additionally, HSC niche maintenance relies on osteolineage, adipocytes and macrophages [20,21,22]. A detailed characterization of the other bone marrow niche components can be found in dedicated reviews [23,24].

## 3. Circular RNAs and Their Role in Hematopoiesis

The hematopoietic process is tightly regulated through gene expression modulation in HSCs and progenitor cells. It is well known that diverse differentiation programs are triggered by both specific transcription factors and non-coding RNAs, such as long non-coding RNAs (lncRNAs) and microRNAs (miRNAs) [25,26,27]. However, there is a need to better understand the roles played by circular RNAs (circRNAs) in hematopoiesis. circRNAs are covalently closed single-stranded molecules derived from back-splicing, non-canonical splicing in which flanking regions of one or multiple exons are joined together. Previously thought to result from random splicing errors, circRNAs were proven to be conserved and associated with inverted repeated Alu sequences within the flanking introns [28]. RNA splicing occurs within the nucleus. During back splicing, the canonical spliceosomal machinery recognizes the splice sites, and circularization can be direct or preceded by the production of a long lariat. The entire process is usually regulated by RNA-binding proteins. For instance, during epithelial to mesenchymal transition, the RNA-binding protein Quaking induces the formation of numerous circRNAs [29]. Human circRNAs are co-transcriptionally or post-transcriptionally produced, most of them containing two or three exons. The vast majority of circRNAs have no introns and undergo nuclear export (Figure 1A). Notably, circularization competes with linear splicing, strongly suggesting that circRNAs have a role in gene regulation [30]. Indeed, several biological functions of circRNAs have been identified so far: they can act as “sponges” for miRNA or RNA-binding proteins, activate gene transcription and, in some cases, they are also translated into proteins in a N 6-methyladenosine (m^6^A)-dependent manner (Figure 1B) [31]. An interesting work by Pengyan Xia and colleagues showed that maintenance of LT-HSCs homeostasis is regulated by a circRNA named cia-cGAS. It binds the DNA sensor cGAS in the nucleus, thus blocking its synthase activity. Indeed, circRNA cia-cGAS deficiency leads to increased levels of type I IFNs and LT-HSCs cycling. Therefore, since cia-cGAS is highly expressed in dormant HSCs and protects them from c-GAS-mediated exhaustion, it is a key factor in keeping the balance between HSCs’ quiescent and cycling state [32]. Nicolet and colleagues identified more than 59,000 circRNAs in hematopoietic cells, providing the first comprehensive analysis of circRNAs expression in these cells [33]. Importantly, they found that circRNAs are cell-type specific, and their expression levels are altered during terminal hematopoietic differentiation. Differentiated cells, especially erythrocytes and platelets, produce more circRNAs than progenitor cells. Just recently, this group also showed that there is limited correspondence between circRNAs expressed in mature cells and those found in their progenitor cells, suggesting that differential expression of circRNAs is a regulated process rather than mere accumulation. Importantly, by comparing the expression of circRNAs with the translation efficiency of the counterpart mRNA, they found a correlation only in a small percentage (0.04%) of cases. Hence, the ways through which these molecules are regulated still need to be elucidated [34]. Nevertheless, these findings indicate a remarkable role for circRNAs in the modulation of hematopoietic differentiation. 

## 4. AML and the Leukemic BM Niche

The starting point of AML onset is the progressive addition of several leukemia-associated mutations in HSCs, which become pre-leukemic HSCs (pre-LSCs) [35]. The conversion from pre-LSCs to fully transformed leukemic stem cells (LSCs) or leukemia-initiating cells (LICs) is a multistep process in which sequential aberrations in transcription, epigenetic regulation and expression of metabolic factors are acquired over the years [36]. While both pre-LSCs and LSCs maintain their self-renewal capacity, they also generate leukemic blasts, which form the bulk of the tumor [37]. In addition to the study of genetic alterations in HSCs resulting in leukemogenesis, a new perspective focusing on alterations of the bone marrow niche has been emerging in the last few years. Indeed, germline mutations in stromal cells can promote dysregulated hematopoiesis, and they are sufficient for the development of AML [38]. Although niche genetic alterations that predispose to hematological malignancies were identified, these studies have been held in mouse models, and the mechanisms occurring in patients need to be elucidated. On the other hand, both leukemia progression and development of therapy resistance occur through a deep remodeling of the bone marrow microenvironment (Figure 2). For instance, neo-angiogenesis is driven by enhanced expression of vascular endothelial growth factor (VEGF) and angiopoietin in both leukemic and bone marrow stromal cells [39,40]. Moreover, patient-derived xenografts (PDX) have a remarkable vascular leakiness, and this abnormal permeability strongly affects drug delivery, proving to be associated with poor prognosis in AML patients [41]. Another strategy used by leukemic cells to invade the bone marrow microenvironment is the disruption of healthy hematopoietic stem and progenitor cells reservoir [42]. Moreover, immune escape is a critical point for leukemia progression and therapy resistance. To this aim, AML blasts implement several strategies: they downregulate surface membrane MHC class I and II molecules to avoid immune recognition, induce NK and T cells dysfunction, favor immunosuppressive Treg cells and alter cytokine milieu in a pro-leukemic way. Of note, mesenchymal stem cells secrete factors so as to foster immunosuppression and recruit Treg and M2 macrophages [43,44]. Compared to their normal counterpart, leukemic cells have an altered energy metabolism, with higher mitochondrial mass and oxygen consumption but no concomitant increase in respiratory chain complex activity. Since AML blasts strongly rely on fatty acid oxidation (FAO) and oxidative phosphorylation (OXPHOS), they are susceptible to oxidative metabolic stress [45,46]. Likewise, some solid tumors and AML cells exploit adipocytes as a source of fatty acids (FA) [47]. Moreover, it was demonstrated that an increase in AML mitochondrial mass is due to the direct uptake of functional mitochondria from bone marrow stromal cells. This mitochondria transfer, which increases ATP production in leukemic blasts, is remarkably enhanced during chemotherapy and confers chemoresistance [48,49,50]. Within the bone marrow, exosomes play a critical role in cell-to-cell communication and in AML-mediated microenvironment remodeling in order to create a self-strengthening leukemic niche [51,52]. These vesicles transfer different bioactive molecules such as proteins, lipids, DNA and RNA. Since hypoxia, nutrients deprivation and acidosis render bone marrow an extremely hostile environment even for leukemic blasts, they cope with ER stress conditions by activating the unfolded protein response (UPR) [53] and it was shown that AML extracellular vesicles (AML-EVs) transfer ER stress to bone marrow mesenchymal stem cells, thus creating a leukemia permissive microenvironment [54]. Ultimately, the anchorage of leukemic stem cells to the niche also plays an essential role in AML pathogenesis [55,56].

## 5. circRNA in AML Pathogenesis

Several circRNAs were found to play oncogenic or tumor-suppressive roles, and aberrant expression was detected in both solid and hematopoietic cancers [57], even though, in many cases, the underlying mechanisms and regulatory networks are still to be clarified. For example, it was shown that N6-methyladenosine (m^6^A) modification has an important regulatory function for circRNAs in AML [58]. Table 1 shows the annotated circRNAs implicated in AML so far. The vast majority of these circRNAs act as miRNA sponges; they are defined as competing for endogenous RNAs (ceRNAs) that, by binding miRNAs, compete for their binding with the target mRNAs and hinder their regulatory function. Therefore, circRNAs can alter gene expression regulation, hence contributing to AML pathogenesis. 

AML is characterized by both chromosomal rearrangements and gene mutations [100]. Translocations are very often present, leading to the production of mutant fusion proteins. An intriguing work by Guarnerio et al. showed that the well-established translocations forming the oncoproteins PML-RARA and MLL-AF9 also produce fusion circRNAs (f-circRNAs), named f-circPR and f-circM9, respectively [59]. These f-circRNAs, derived from the aberrant conjunction of exons from different genes, contribute to HSCs transformation, leukemia progression and therapy resistance. Regarding genetic mutations, the most common genetic lesion in AML affects the gene encoding for nucleophosmin-1 (NPM1). Mutated NPM1 protein loses its nucleolar localization, and it is delocalized in the cytoplasm, where it is responsible for the block of myeloid cell differentiation [101]. There is a circRNA encoded by the NPM1 gene, circNPM1, that is highly expressed by AML patients and cell lines and is associated with lower expression of members of the Toll-Like Receptors family, which are involved in normal hematopoietic differentiation. The effects of circNPM1 on TLR pathway genes could be mediated through miR181 [61]. Moreover, another study showed that circNPM1 silencing counteracted AML chemoresistance to Adriamycin. circNPM1 is a ceRNA for miR-345-5p, leading to increased expression of the miR-345-5p target gene FZD. Notably, FZD5 is an oncogene in various cancers. Since circNPM1 serum levels are high in AML patients, it might be a potential biomarker for drug resistance in AML [62]. Another protein frequently mutated in AML is the tyrosine kinase receptor FLTInternal Tandem Duplications of its juxtamembrane domain generate the oncoprotein FLT3-ITD, which is associated with very poor prognosis [102]. It was demonstrated that circMYBL2—derived from cell cycle checkpoint gene MYBL2—is upregulated in AML patients carrying FLT3-ITD mutation. This circRNA is crucial in promoting FLT3 mRNA translation by recruiting the RNA binding protein PTBP. Moreover, circMYBL2 silencing impairs leukemic cells proliferation in vivo and overcomes acquired resistance to quizartinib. Therefore, circMYBL2 may be a potential therapeutic target for FLT3-ITD AML patients [72]. Another interesting example is circPAN3, deriving from the gene encoding for PAN3 exonuclease. This circRNA is highly expressed in both AML cell lines and primary blasts resistant to doxorubicin. It promotes autophagy through the AMPK/mTOR pathway, thus conferring drug resistance [67]. Autophagy was found to be a mechanism for resistance in a range of solid tumors. Indeed, circPAN3 silencing can reduce autophagy and restore drug sensitivity. Moreover, an additional study showed that circPAN3 action might depend on miR-153-5p/miR-183-5p-XIAP axis [68]. XIAP is an anti-apoptotic protein that binds caspases 3, 7 and 9, leading them to proteasome-mediated degradation. However, circPAN3 downregulation decreases XIAP levels, and this could be due to sponging activity on miR-153-5p and miR-183-5p. These findings confirm that circPAN3 could be a predictor for treatment efficacy and also a therapeutic target in chemoresistance. Another circRNA related to AML originates from the gene RNF220, encoding for a RING domain E3 ubiquitin ligase that mediates ubiquitination of multiple targets. circRNF220 is especially expressed in pediatric AML, and it is a predictor of relapse. It acts as a sponge for miR-30a, thus increasing levels of its target mRNAs, including MYSM1 and IER. Interestingly, MYSM1 is a key transcription factor in hematopoiesis, while IER2 is upregulated in many tumors and is associated with cancer progression and metastasis. Therefore, circRNF220 could be useful as a prognostic marker, particularly in terms of relapse prediction [77] (Figure 1C).

## 6. circRNA in Leukemic Bone Marrow Niche

Although some circRNAs identified so far definitely lack a full description of the molecular mechanisms that render them pro-oncogenic or tumor-suppressive molecules, their association with AML development, progression and relapse has been proven (see Table 1). It is currently not clear whether these circRNAs are directly involved in the interplay between AML and the bone marrow microenvironment, but since leukemic blasts deeply change the composition of the niche to their advantage, the idea that circRNAs may be a part of this cell–cell communication is definitely intriguing. In support of this, enrichment of circRNAs in exosomes was reported in various diseases, including cancer [103]. In different types of solid tumors, it was demonstrated that circRNAs have a prominent role in regulating cancer cell metabolism, in particular, glycolysis, fatty acid oxidation, oxidative respiration and glutamine production [104]. Moreover, it is known that some circRNAs strongly interact with the tumor microenvironment in order to promote different steps of metastasis, including cancer cell migration, invasion, intravasation and neo-angiogenesis [105]. There is a reason to think that circRNAs involved in these processes may be packaged in exosomes. In regard to AML, the key role of extracellular vesicles in implementing bone marrow niche remodeling is well-established. It was shown that levels of plasma-derived exosomes were higher in newly diagnosed AML patients and that they contained a different cargo compared to normal cell-derived exosomes. Notably, the exosome amount was decreased during remission [106]. It is noteworthy that hsa_circ_0009910 was found upregulated in AML cells and especially in AML-derived exosomes. This circRNA exerts an oncogenic role by acting as a miRNA sponge in the miR-5195-3p-GRB10 axis, in which GRB10 is an adapter protein that is involved in aberrant proliferation in FLT3-ITD positive AML. The authors proposed that hsa_circ_0009910 may be shuttled via exosomes to surrounding AML cells in order to promote their malignant properties [73]. In view of the evidence concerning AML resistance mechanisms uncovered so far, the role of circRNAs in intercellular communication within the leukemic bone marrow niche needs to be further investigated. These studies could provide insights into new therapeutic opportunities aimed at avoiding relapse of the disease. Moreover, in addition to the accumulation within the bone marrow, in some cases, leukemic cells can infiltrate other organs. This phenomenon is called Extramedullary Infiltration (EMI), and it is quite common, with myeloid sarcoma and leukemia cutis appearing in 1.4–9% and 15% of AML patients, respectively [107]. It is known that EMI is often associated with poor prognosis and relapse/refractory AML. An interesting paper shows that EMI and non-EMI AML samples have a different circRNA/miRNA/gene regulatory network, with circRNAs in EMI involved in cell adhesion, migration, signal transduction and cell–cell communication [90]. In particular, hsa_circ_0004520, which increases PLXNB and VEGFA levels, could promote angiogenesis. 

## 7. Conclusions 

The list of circRNAs involved in AML has been increasing fast in the very last few years. This is promising in the view of new therapeutic approaches, but it is essential to deal with some critical limitations. A complete and updated database and a unified nomenclature for circRNAs should be generated to avoid confusion in their classification. CircBase is a useful tool in which merged data sets of circRNAs are freely accessible [108]. Other strong bioinformatics tools are circIMPACT and CRAFT, which are able to identify regulatory networks governed by circRNAs and produce functional predictions [109,110]. From present studies, several circRNAs could be potential diagnostic or prognostic biomarkers for AML. Their particular structure renders them much more stable than their linear cognate RNA. Their presence both in the bone marrow and in body fluids, such as blood and urine, is also a remarkable point, supporting their use as biomarkers. Regarding circRNAs mechanisms of action, the vast majority of them have been shown to act as miRNA sponges. However, the assessment of copy number and of the number of miRNA binding sites is essential for carrying out these functional studies. Indeed, a rare circRNA with many miRNA binding sites could be equally effective as miRNA sponges as an abundant one with few miRNA binding sites. In order to measure the copy number of circRNAs with high accuracy, quantitative PCR or digital PCR should be used. Moreover, the assessment of diagnostic potential through specific tests is another important point [111]. Regarding upregulated circRNAs in AML and their application as therapeutic targets, the use of antisense oligomers (ASOs) is a valuable option. However, ASOs need to target the junction sequence, specific for circRNA; otherwise, parental linear RNA silencing occurs as a side effect. An interesting alternative to avoid off-target effects could be the use of circular siRNAs [112]. For downregulated circRNAs, overexpression could be obtained through exogenous delivery methods such as nanocarriers and nanoparticles. A novel method based on ferritin nanoparticles, which deliver nucleic acids specifically into AML cells, was recently developed [113].

The study of pro-survival strategies implemented by AML during both progression and development of therapy resistance is tangled. Scientists have only recently approached the investigation of altered AML niche as a mine of information about leukemia necessities and vulnerabilities. Crosstalk between AML and the bone marrow microenvironment occurs through a complex mutual exchange of molecular signals, hence establishing a symbiotic relationship that allows disease progression and chemoresistance. circRNAs are likely to be involved in this process. Deep knowledge of circRNAs transport within and outside the cell is necessary in order to clarify their involvement in cell–cell communication. Future evaluation of circRNAs biological mechanisms will shed light on new promising strategies for AML treatment.

## Figures and Tables

**Figure 1 ncrna-08-00050-f001:**
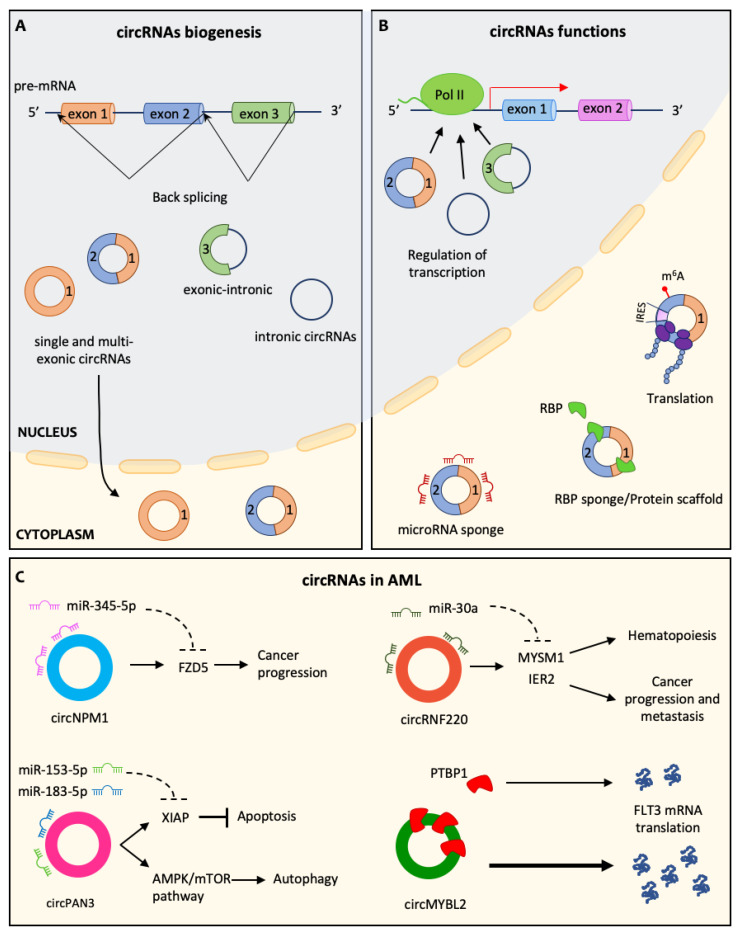
(**A**) circRNAs biogenesis. circRNAs are generated by unconventional splicing, named back-splicing, in which the 3’ end of an exon binds the 5’ end of an upstream exon, forming a covalently closed molecule. circRNAs can be divided into three main classes: single or multi-exonic circRNAs (on the left), which can translocate in the cytoplasm; exonic-intronic circRNAs (in the middle) and intronic-circRNAs (on the right), which are retained in the nucleus. (**B**) circRNAs functions. CircRNAs can regulate transcription, interacting with Polymerase-II (Pol II) in the nucleus. In the cytoplasm, they can act as microRNA sponges or RBP sponges, and they can also be translated thanks to the presence of an IRES or m6A modification. (**C**) circRNAs in AML. Representation of some circRNAs involved in AML progression: circNPM1 acts as miR-345–5p sponge, increasing FZD5 expression, a well-known oncogene; circPAN3, sponging miR-153-5p and miR-183-5p, leads to increased autophagy and inhibits apoptosis; circRNF220 induces MYSM1 and IER2, affecting hematopoiesis and promoting cancer progression and metastasis; circMYBL2 favors mRNA FLT3 translation by binding PTBP1.

**Figure 2 ncrna-08-00050-f002:**
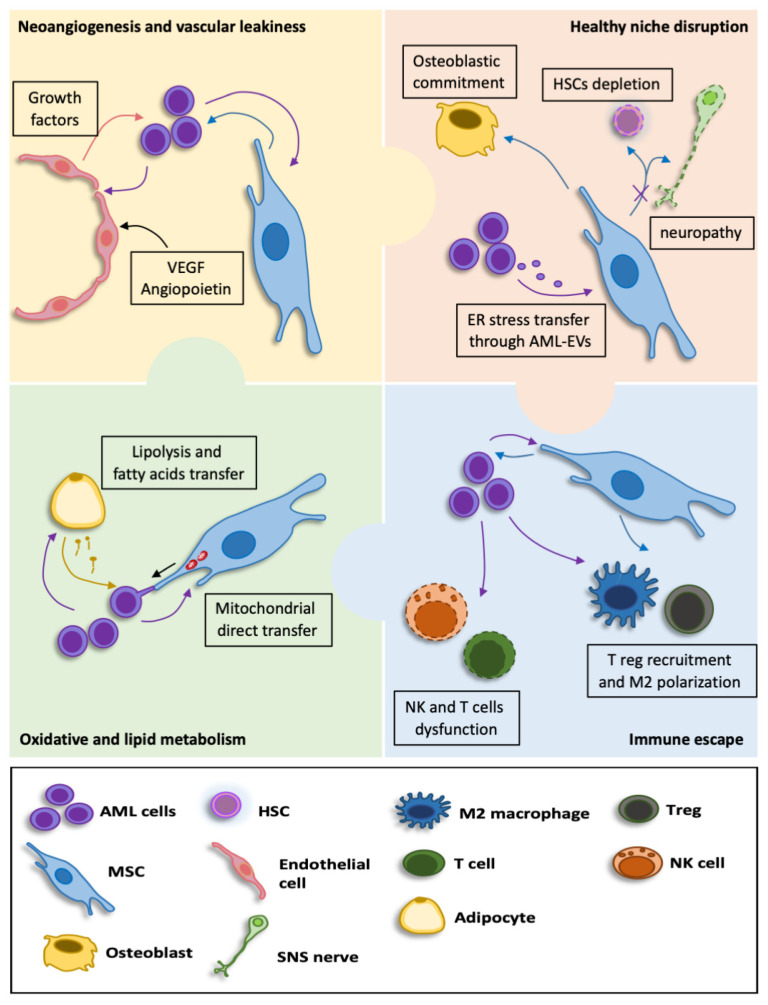
Interaction between AML cells and the bone marrow microenvironment. Within the niche, AML cells communicate with mesenchymal stem cells and endothelial cells in order to increase bone marrow vascularization and alter vascular permeability (**yellow panel**); Moreover, AML cells strongly reprogram MSCs, leading to a self-reinforcing niche at the expense of normal hematopoiesis (**pink panel**); Leukemic cells escape the immune system and recruit anti-inflammatory components such as Treg and M2 macrophages (**light blue panel**); AML cells exploit stromal cells to enhance their antioxidant defenses and adipocytes as a source of cellular energy (**green panel**).

**Table 1 ncrna-08-00050-t001:** Updated list of circRNAs involved in AML onset, progression and therapy resistance. Arrows **↑** and **↓** stand for upregulation and downregulation respectively.

Name	Gene of Origin	Levels	Pathway Targeted/ Mode of Action	Impact	Ref.
f-circPR	PML-RARA	de novo in AML	AKT signaling	Increased cell proliferation and chemotherapy resistance	[59]
f-circM9	MLL-AF9	de novo in AML	MAPK and AKT signalling	Increased cell proliferation and chemotherapy resistance	[59]
circ ANAPC7 hsa_circ_101141	ANAPC7	↑ in AML	miR181	Prognostic biomarker	[60]
circNPM1 hsa_circ_0075001	NPM1	↑ in AML	mir181 and TLR signalling miR-345-5p/FZD5	Hematopoietic differentiation Chemotherapy resistance	[61,62]
circDLEU hsa_circ_0000488	DLEU	↑ in AML	miR496/PRKACB	Increased cell proliferation and apoptosis inhibition	[63]
circANXA2 hsa_circ_0035559	Annexin A2	↑ in AML	miR-23a-5p and miR-503-3p	Prognostic biomarker and chemotherapy resistance	[64]
circVIM	Vimentin	↑ in AML	Unknown	Diagnostic and prognostic biomarker	[65]
circHIPK2	HIPK2	↓ in AML (APL)	miR-124-3p/CEBPA	Prognostic biomarker and ATRA-induced differentiation	[66]
circPAN3 hsa_circ_0100181	PAN3	↑ in AML ADM resistant	AMPK/mTOR miR-153-5p/XIAP	Chemotherapy resistance	[67,68]
hsa_circ_0004277	WDR7	↓ in AML	miR-134-5p/SSBP2	Diagnostic and prognostic biomarker	[69,70]
hsa_circ_0003602	SMARCC1	↑ in AML	miR-502-5p/IGF1R	Increased cell proliferation and apoptosis inhibition	[71]
circMYBL2 hsa_circ_0006332	MYBL2	↑ in AML FLT3-ITD+	PTPB1/FLT3 translation	Increased cell proliferation and resistance to quizartinib	[72]
hsa_circ_0009910	MFN2	↑ in AML and AML exosomes	miR-5195-3p/GRB10 miR-20a-5p	Increased cell proliferation and apoptosis inhibition	[73,74]
hsa_circ_0121582	GSK3beta	↓ in AML	miR-224/GSK3β/Wnt/βcatenin	Inhibited cell proliferation	[75]
circFOXO3	FOXO3	↓ in AML	Apoptotic pathways	Increased apoptosis Diagnostic and prognostic biomarker	[76]
circRNF220 hsa_circ_0012152	RNF220	↑ in AML relapse	miR30a/MYSM1-IER2	Increased cell proliferation and apoptosis inhibition, biomarker to predict relapse	[77]
hsa_circ_100290	SLC30A7	↑ in AML	miR-203/Rab10	Increased cell proliferation and apoptosis inhibition	[78]
circRNF13 hsa_circ_0001346	RNF13	↑ in AML	miR-1224-5p	Increased cell proliferation and apoptosis inhibition	[79]
hsa_circ_104700	PTK2	↑ in AML	miR-330-5p/FOXM1	Increased cell proliferation and apoptosis inhibition	[80]
hsa_circ_002483	PTK2	↑ in AML	miR-758-3p/MYC	Increased cell proliferation and apoptosis inhibition	[81]
hsa_circ_0005774	CDK1	↑ in AML	miR192-5p/ULK1	Increased cell proliferation and apoptosis inhibition	[82]
circCRKL	CRKL	↓ in AML	miR-196a-5p/p27 miR-196b-5p/p27	Inhibited cell proliferation	[83]
circPOLA2	POLA2	↑ in AML	miR-34a	Increased cell proliferation	[84]
hsa_circ_0079480	ISPD	↑ in AML	miR-654-3p/HDGF	Increased cell proliferation and apoptosis inhibition Prognostic biomarker	[85,86]
circKLHL8	KLHL8	Associated with outcome	miR-155/CDKN1-CDKN2-BCL6-TLR4-CEBPD-CEBPB	Prognostic biomarker	[87]
circFBXW7	FBXW7	↓ in AML	Signal transduction Leukocyte differentiation	Tumor suppressor	[87]
circ_KCNQ5 hsa_circ_0004136	KCNQ5	↑ in AML	miR-142 miR-622/RAB10	Increased cell proliferation and apoptosis inhibition	[88,89]
hsa_circ_0004520	VAV2	↑ in AML	PLXNB2, VEGFA	Angiogenesis Prognostic biomarker for EMI	[90]
hsa_circ_0000370	FLI-1	↑ in AML FLT3-ITD+	miR-1299/S100A7A	Prognostic biomarker	[91]
circ_0040823	BANP	↓ in AML	miR-516b/PTEN	Inhibited cell proliferation and increased apoptosis	[92]
circPLXNB2	PLXNB2	↑ in AML	PLXNB2	Increased cell proliferation and migration, apoptosis inhibition Prognostic biomarker for EMI	[93]
circ_0002232	PTEN	↓ in AML	miR-92a-3p/PTEN	Diagnostic and prognostic biomarker	[94]
circ_0012152	RNF220	↑ in AML	miR-625-5p/SOX12	Increased cell proliferation and apoptosis inhibition	[95]
circ_SFMBT2 hsa_circ_0017639	SFMBT2	↑ in AML	miR-582-3p/ZBTB20	Increased cell proliferation, migration and invasion	[96]
circ-PVT1	PVT1	↑ in AML	c-Myc and BCL-2?	Prognostic biomarker	[97]
hsa_circ_0001947	AFF2	↓ in AML	miR-329-5p/CREBRF	Inhibited cell proliferation Prognostic biomarker	[98]
hsa_circ_0075451	GMDS	↑ in AML	miR-330-5p/PRDM16 miR-326/PRDM16	Prognostic biomarker	[99]

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
