# Peer review of "Circular RNAs Activity in the Leukemic Bone Marrow Microenvironment"

_ncrna, 2022, doi:10.3390/ncrna8040050_

Round 1

Reviewer 1 Report

 In this manuscript, authors provide an in-depth review of the role of circular RNAs activity in the leukemic bone marrow microenvironment. Disease recurrence and poor outcomes remain a central issue in AML (Acute Myeloid Leukemia) management despite the recent approvals of several targeted therapies. Bone Marrow (BM) niche remodeling and a complex crosstalk between AML cells with its microenvironment has been known to affect therapy responsiveness. Authors highlight the role of recently discovered class of RNA molecules called circRNAs and their potential contribution to cell-cell communication within the leukemic BM microenvironment and stipulate that a better understanding of these mechanisms will help develop more effective treatment strategies for AML.

Points:

1.     Line 29-31: Meaning of this statement is not clear. Do the authors mean to say that molecular heterogeneity is a major contributing factor in its diverse presentation, symptoms, prognosis, and responsiveness to treatment? Rephrasing this sentence could make it more concise.

2.     Line 58-127: In this section, authors describe in details the hematopoietic stem cells and their niche including a detailed account of their regenerative hierarchy. While this is great description, I feel it’s too long, authors can simply include a graphic depicting the same with couple of lines highlighting the features. The focus of the reviews is on circRNAs, and this takes away from the message.

3.     Line 132-133: Meaning not clear, do authors mean to say the “There is a need to better understand the roles played by circRNAs in hematopoiesis ” ?

4.     Throughout the manuscript there are many grammatical errors as well as errors in sentence construction which are affecting the message its supposed to convey. Authors need to carefully revisit the text and fix those errors (highlighting some of those in the comments).

5.     Line 12: Usage of the word “In spite” or something similar would convey the message better than the word “Nonetheless” in this context.

6.     Line 26: spelling mistake: “un” instead of “in”

7.     Line 60: “endowed with” instead of “endowed of”

8.     Page 6: Figure 1: Figure legends under the panel can be in the same width as the main panel. Also, the text needs to be justified.

9.     Line 255-256:  “sentence ending in “at the very beginning”” needs to be rephrased. Additionally, “Table 1 shows the circRNAs implicated in AML annotated so far” needs to be corrected to read “Table 1 shows the annotated circRNAs implicated in AML so far”

10.  Line 311: What does this mean “In different solid tumors”, do authors mean in different cancer entities?

11.  Line 317: What does this mean “As concerns AML”, do authors mean “In regard to AML”?

12.  Line 327: “mechanisms obtained so far”, do authors mean “mechanisms uncovered so far”?

13. For this review to appeal to a wide audience, it would be great if authors consider structuring it centering around circRNAs, may be provide a graphic for biogenesis of circRNAs, known mechanism of actions and then focus on AML and BM associated circRNAs. 

Summary:

Authors summarize and review BM niche as well as provide a updated list of known circRNAs in AML. CircRNAs are a promising class of biomolecules with potential in early diagnosis, predicting treatment outcomes and as well as in providing better therapeutic interventions. This review in its current form seems to focus more on history of AML and BM and very little on circRNAs and the known mechanisms. In my opinion, there are multiple reviews and studies online doing exactly that. This review would be more useful to the scientific community if the structure of this manuscript is altered to 1 page max for AML and BM history whereas rest discussing and describing in details circRNAs currently known and providing more context to their discovery in terms of pathways / cancer hallmarks they are regulating.

Reviewer 2 Report

AML is the disease of haematopoetic stem cells in the bone marrow. The authors analyze the role of circular RNAs in the bone marrow microenvironment. Though there are different circular RNAs in the bone marrow, there are some points that could be improved in the review. 

How circular RNAs could be used as treatment biomarkers could be discussed in a few lines?

How therapeutically relevant are circular RNAs and are they present in normal CD34+ cells?

Few places, there are some typos such as NPM1..(NMP1) is written. This could be corrected. 

Author Response

AML is the disease of haematopoetic stem cells in the bone marrow. The authors analyze the role of circular RNAs in the bone marrow microenvironment. Though there are different circular RNAs in the bone marrow, there are some points that could be improved in the review.

How circular RNAs could be used as treatment biomarkers could be discussed in a few lines?

Thank you for your advice. Several circRNAs identified so far could be used as biomarkers. We have specified this concept in the paragraph 5 “circRNA in AML pathogenesis”, in which we have provided some examples of circRNAs involved in AML. The entire list of circRNAs annotated as useful biomarker in AML can be read in Table 1 (see “Impact” column).

How therapeutically relevant are circular RNAs and are they present in normal CD34+ cells?

circRNAs therapeutic relevance is discussed in both paragraph 5 “circRNA in AML pathogenesis” and in paragraph 7 “Conclusion”. They are present in normal CD34+ hematopoietic stem cells, as discussed in paragraph 3 “Circular RNAs and their role in hematopoiesis”. We have also provided the example of cia-cGAS, a circRNA that is essential for hematopoietic stem cells maintenance.

Few places, there are some typos such as NPM1..(NMP1) is written. This could be corrected.

We are sorry about that, thank you for your advice. The abbreviation has been corrected and the text has been carefully revised for mistakes and typos.

Round 2

Reviewer 1 Report

Authors have answered all the points initially raised.